# Copolymerization of Styrene and Pentadecylphenylmethacrylate (PDPMA): Synthesis, Characterization, Thermomechanical and Adhesion Properties

**DOI:** 10.3390/polym12010097

**Published:** 2020-01-04

**Authors:** Tomy Muringayil Joseph, Sumi Murali Nair, Suresh Kattimuttathu Ittara, Józef T. Haponiuk, Sabu Thomas

**Affiliations:** 1Polymers Technology Department, Chemical Faculty, Gdansk University of Technology, 80-233 Gdansk, Poland; 2Polymers & Functional Materials Division, CSIR-Indian Institute of Chemical Technology, Hyderabad 500007, India; sumimuralinair@gmail.com (S.M.N.); kisuresh@iict.res.in (S.K.I.); 3School of Chemical Sciences, Mahatma Gandhi University, Kottayam 686560, India; sabuchathukulam@yahoo.co.uk

**Keywords:** copolymerization, bioderived monomer, reactivity ratio, molecular weight, polydispersity, adhesion properties

## Abstract

The copolymerization of styrene (St) with a bioderived monomer, pentadecylphenyl methacrylate (PDPMA), via atom transfer radical polymerization (ATRP) was studied in this work. The copolymerization reactivity ratio was calculated using the composition data obtained from ^1^H NMR spectroscopy, applying Kelen-Tudos and Finemann-Ross methods. The reactivity ratio of styrene (r_1_ = 0.93) and PDPMA (r_2_ = 0.05) suggested random copolymerization of the two monomers with alternation. The copolymerization conversion increased with increasing PDPMA concentration of the feed, upto 70 wt % PDPMA, but decreased thereafter. The molecular weight determined by gel permeation chromatography was lower than the theoretical values and the polydispersity increased from 1.32 to 2.19, with increasing PDPMA content in the feed. The influence of styrene content on the glass transition and thermal decomposition behavior of the copolymers was studied by differential scanning calorimetry (DSC) and thermogravimetric analysis, respectively. Morphological characterization by transmission electron microscopy (TEM) revealed a phase separated soft core-hard shell type structure. The complex viscosity and adhesion properties like peel strength and lap shear strength of the copolymer on different substrates increased with increasing styrene content.

## 1. Introduction

Bio-based materials decrease the ecological footprint at every single phases of the product’s life cycle. Global spread of economic development and rapidly rising population are highly demanding towards the world’s resources. The textile, food and plastic packaging industry have attained greater exposure in the core of the sustainability agenda as consumers are directly affected by its utilization. Recycling of bio-based materials should be more based on composting as a terminating procedure and then such products will become more attractive. The process development to tailor the physico-chemical properties of bio-based materials to specific end application is a significant tool, like cross-linking the matrix to enhance mechanical strength. It is worth noticing that even if at the current stage, such materials seem to be a tiny compromise between economic efficiency and sustainability. Recent advances focus on the development of pure bio-based products identical to the petroleum-based materials.

There are many reports on the development of sustainable products using biobased materials and their structure-property correlation. Alginate fibres are used in commercial wound dressings as they resorb wound exudate while even a fence to external infective agents assisting a suitable microenvironment for tissue regrowth [1]. Chitosan, being biodegradable, non-toxic and biocompatible is presented in reports as a candidate for several applications in various industrial products like those from biomedical, pharmaceutical, agricultural, food, cosmetics, and textiles industries [1]. The development of films and coatings by proteins for applications in paper products or board, besides, prospective industrial potential also has been reported. The amide-based chemical structure of proteins shows diverse properties however it is the most significant fact that the high intermolecular binding potential such that protein-based films exhibit improved mechanical properties compared to fat-based films or polysaccharide films [2]. The multifunctional effect could be achieved from green coconut shell extract (CSE) employed on wool (proteinous), jute (lingo-cellulosic) and cotton (cellulosic) that can be a worthy low-cost bio-based packaging material [3,4,5]. Another review reports the properties of sugar palm fibres and starch and its products in green composites and also revealed the prospects of sugar palm fibres and biopolymer for industrial applications such as packaging, bioenergy, automotive and others [6]. The cost-efficient films thus manufactured would have applications where biodegradable films are usable for packaging films, consumer bags, agricultural mulch films and silage wraps [7].

Biobased monomers and polymers, in-lieu of their structural diversity and complexity, offer tremendous scope for sustainable material development through appropriate chemical modification strategies [8,9,10,11,12,13,14,15,16]. Biobased alkyl (meth) acrylate monomers, with long alkyl side chain like lauryl, stearyl or octyl chain, yield polymers with special properties [16,17,18]. The basic and applied research potential of several of these resources has been well reported [19,20,21,22,23,24,25,26,27,28,29]. The effect of polymer synthesis conditions and techniques employed on the polymer as well as application properties was studied by several groups [23,24,25,26,27,28,29].

Cardanol, obtained from cashew nut shell liquid, is an important phenolic renewable resource carrying a 15-carbon side chain in the meta position of the benzene ring with varying degree of unsaturation and useful in the development of value added products [30,31]. It has been utilized in several industrial applications, ranging from adhesives and coatings, to lining materials, and recently, the development of some new polymers with novel properties have been reported [32,33,34,35]. We have reported the application of controlled radical polymerization methods, like atom transfer radical polymerization (ATRP), to cardanol based monomers which yielded polymers with novel processability, performance, and its nanocomposite with graphene showing superior adhesion and surfactant properties [36,37,38]. In this regard, the acrylate and methacrylate derivatives of hydrogenated cardanol (PDP, 3-pentadecylphenol) offer great potential and consistent characteristics as compared to cardanol. The complexation of PDP with block copolymers and its side chain crystallization was studied by ten Brinke et al. and reported the formation of comb-like polymers resulting from immiscibility between alkyl side chain and the main chain [39,40,41,42]. The synthesis of a novel reactive surfactant starting from the acrylate derivative of PDP and its application in the preparation of internally plasticized polystyrene with improved hydrophobic characteristics was reported by us [43,44]. The methacrylate derivative of PDP was hybridized with graphene and the nanocomposite showed side chain crystallization and improved adhesion properties [37]. The adhesion properties could be further controlled by copolymerization with monomers like styrene or MMA and the copolymerization data are important in such cases [45]. In this work, we report the copolymerization of styrene with pentadecylphenylmethacrylate (PDPMA) via ATRP and the effect of styrene content on the conversion, molecular weight, and polydispersity was studied. From the composition data the reactivity ratios were calculated using Finemann Ross and Kelen-Tudos methods. The side chain crystallization behavior and thermal stability of the copolymers were studied using DSC and TGA, respectively. The adhesion, morphology and rheological properties have also been reported.

## 2. Materials and Methods

### 2.1. Materials

The monomers, styrene (St) and methacrylic acid (MAA), as well as the ligands, pentamethyldiethylenetriamine (PMDETA) and bypyridine (dNBpy), initiator ethyl-2-bromo isobutyrate (EBriB), metal catalyst Cu(I)Br, ethanol, glacial acetic acid and diethyl ether were purchased from Sigma Aldrich, St. Louis, MO, USA and used as such. Hydrogenated cardanol (pentadecylphenol, PDP) was kindly supplied by Cardolite Specialty Chemical LLP, Mangaluru, India. Methacryloyl chloride was prepared from methacrylic acid and benzoyl chloride which are purchased from Sigma-Aldrich, St. Louis, MO, USA and used as such. Triethylamine (TEA) was purchased from Spectrochem, Mumbai, India. Styrene monomer was purified by washing with aqueous (5%) NaOH solution, deionised water and finally distilled under reduced pressure and stored at −18 °C. The metal catalyst, Cu(I)Br, was purified by stirring with glacial acetic acid overnight, washed with ethanol and diethyl ether, filtered and dried at 50 °C under vacuum for 6 h.

### 2.2. Monomer Synthesis

Pentadecylphenylmethacrylate monomer was prepared from methacryloyl chloride and pentadecylphenol as per Scheme 1, according to the procedure of Sitaramam and Chatterjee [28].

In a typical experiment 125 mL dichloromethane and 50 gm (0.16 mol) pentadecylphenol (PDP) were taken in a 250 mL 3-neck RB flask and kept under magnetic stirring. The homogeneous solution was cooled to 0–5 °C and 23 mL (0.17 mol) of triethyl amine was added slowly. Then, 15.6 mL (0.16 mol) of methacryloyl chloride was added slowly over a period of 1 h. The reaction was continued for 12 h at room temperature (RT). The reaction mixture was neutralized with 5% sodium bicarbonate solution and washed with distilled water. The product was extracted in diethyl ether and dried over anhydrous sodium sulphate. The product was further purified on a silica gel column (60–120 mesh) using hexane-ethyl acetate (98:2) solvent system. The monomer structure was confirmed by ^1^H NMR and mass spectroscopy. Product yield was 71%.

### 2.3. Atom Transfer Radical Polymerization (ATRP)

The homopolymerization of PDPMA was carried out using a typical ATRP procedure reported earlier [29], as in Scheme 2 below.

In a typical ATRP experiment (Scheme 2), CuBr (11.46 mg, 0.00008 mol), PMDETA (33.51 µL, 0.00016 mol) and monomer 3-pentadecylphenylmethacrylate (PDPMA) (3 g, 0.008 mol) were added to a two neck flask and magnetically stirred until homogeneous. The reaction mixture was purged with argon through repeated freeze-thaw cycles. Then, ethyl-2-bromo-isobutyrate (11.8 μL, 0.00008 mol) was added as initiator using the syringe and the reaction mixture was kept at 95 °C. The reaction mixture became viscous in about 30 min and the product was diluted with THF (5 mL), and passed through silica gel column (100–200 mesh) for removing the catalyst. The product was purified through repeated precipitation into methanol-water [75:25] mixture, repeated three times, and the final product was dried at 50 °C under vacuum 4 h.

The styrene homopolymer was also prepared following standard procedure. In a typical experiment, CuBr (41.3 mg, 0.00028 mol), PMDETA (119 µL, 0.000576 mol) and monomer styrene (St) (3 g, 0.0288 mol) were taken in a two neck flask and magnetically stirred until homogeneous. The reaction mixture was freeze dried and purged with argongas for 15 min through repeated freeze-thaw cycles. Initiator ethyl-2-bromoisobutyrate (43 μL, 0.00028 mol) was added through a syringe and the flask was kept in a thermostated bath maintained at 95 °C. The polymerization was continued for specified time and the product was diluted with THF (3 mL), passed through a silica gel column to remove the catalyst and then precipitated into methanol-water [75:25] mixture for recovery of the polymer. The procedure is repeated thrice, and then dried at 60 °C under vacuum for 6 h. Conversions were calculated gravimetrically and polymer structure was confirmed by ^1^H NMR.

The copolymerization experiments were carried out at different feed compositions following similar procedure. The required weight of monomers and the catalyst CuBr (0.028 mol) were first charged into a 50 mL RB flask. The ligand PMDETA (119 µL 0.0288 mol) and initiator ethyl-2-bromoisobutyrate (42 μL, 0.0288 mol) were added to flask and sealed. The contents were subjected to repeated freeze thaw cycles (3 times) and sealed under argon. The flask was placed in an oil bath maintained at 95 °C. After 14 h, the reaction was quenched by adding methanol to the reaction mixture. The product was dissolved in THF and precipitated into methanol-water mixture. The procedure was repeated till the isolated polymer was freed from residual monomer, as confirmed by ^1^H NMR spectroscopy.

### 2.4. Characterization

The IR spectrum was recorded using a Thermo Nicolet Nexus 670 spectrometer, (Thermo Fisher Scientific, Waltham, MA, USA). The spectra were recorded at a resolution of 4 cm^−1^ using KBr pellets at room temperature and a minimum of 32 scans were signal averaged. The ^1^H NMR spectra were recorded on an AVANCE-500 (Bruker, Switzerland) spectrometer in CDCl_3_ with tetramethylsilane (TMS) as an internal standard. Mass spectra were recorded on FINNGAN LCQ Advance Max (Thermo Fisher Scientific, Waltham, MA, USA) in methanol. Molecular weights of the synthesized copolymers were determined by gel permeation chromatography (GPC) on a Waters machine (1200 series, Waldbronn, Germany) equipped with a column combination of Styragel HR 5E and Styragel HR 3 (300 mm × 7.8 mm, 6 µm) and a RI detector. Calibration plots were prepared using narrow molecular weight 1, 4-Polyisoprene standards (molecular weights: 31,400−10,500 g/mol from Polymer Labs). THF was used as the eluant at a flow rate of 0.8 mL min^−1^ at 30 °C. Thermal stability studies were performed on a TGA machine from TA Instruments, New Castle, PA, USA (Model: Q500). The tests were run using 3–5 mg samples at 10 °C/min, from RT to 800 °C under nitrogen atmosphere. Thermal transitions were recorded on a TA-Q100 differential scanning calorimeter (DSC) at a heating/cooling rate of 10 °C/min under nitrogen atmosphere. Electron microscopy analysis was performed on samples dispersed in THF:water mixture (9:1) and deposited on carbon coated copper grids and dried. TEM measurements were conducted using a PHILIPS TECHNAI FE12 instrument (Phillips, Holland) at 200 kV. The rheological properties were measured in oscillatory shear mode on a DMA instrument, Anton Paar, Graz, Austria, (model: MCR102).

## 3. Results and Discussion

The objective of the present work was to develop copolymers of PDP methacrylate and evaluate the influence of copolymerization on the adhesion and viscoelastic properties. The adhesion properties of PDPMA and its nanocomposites were reported by us and others previously. The PDPMA monomer was synthesized from hydrogenated cardanol and characterized by FT-IR, ^1^H-NMR and mass spectroscopy. The ^1^H NMR spectrum of PDPMA monomer (Appendix A) revealed its important structural features of the monomer. The characteristic aromatic proton signals appear at 6.9, 7.2 ppm. The terminal methyl (−CH_3_) protons of the alkyl side chain appeared at δ 0.87 ppm. The resonance signal at 2.6 ppm was assigned to the presence of α-CH_2_ protons adjacent to the phenyl ring (Ph−**CH_2_**−(CH_2_)_13_ CH_3_ of PDPMA), in the aliphatic side chain. The vinyl proton signals were seen at 5.6 and 6.4 ppm (multiplet 2H, −C(CH_3_)=CH_2_). The prominent peak at δ 1.25 ppm was assigned to the internal methylene protons (−CH_2_−(CH_2_)_12_−CH_3_) and the singlet observed at 2.1 ppm was assigned to the methacrylate methyl group.

The FT-IR (Appendix A) showed characteristic C−H stretching vibration at about 2926 cm^−1^. The –C=O− group stretching was at 1756–1747 cm^−1^. The −C=C− acrylic and aromatic vibrations were centred at 1645 cm^−1^ and 1629 cm^−1^, respectively. The monomer structure was further confirmed using mass spectroscopy (Appendix A), with the molecular ion peak at 390.6, corresponding to the molar mass of ammonium adduct (*m*/*z* + 18.039 g/mole) and the peak at 373.6 was assigned to the proton adduct (*m*/*z* + 1).

### 3.1. Polymer Synthesis and Characterization

The monomer PDPMA was polymerized using atom transfer radical polymerization (ATRP) and the copolymers with styrene were also prepared. The copolymerization scheme of styrene and PDPMA using bulk ATRP at 95 °C is shown as Scheme 3.

The composition and characteristics of the styrene-PDPMA copolymers synthesized via ATRP are summarized in Table 1.

The homopolymerization of PDPMA was very fast and the reaction mixture became every viscous in about half an hour. However, only 7% conversion was achieved with styrene homopolymerization under the same conditions. Typical ^1^H NMR spectrum of the PDPMA homopolymer synthesized via ATRP is shown in Figure 1. As compared to Figure 1, the main difference observed in ^1^H NMR upon polymerization was disappearance of the vinylic proton peaks.

The copolymerization experiments were run for 14 h to observe viscosity build up, so that appreciable conversion was reached. The copolymer structure was confirmed by ^1^H NMR (Figure 2) and the composition was determined by integration of peaks characteristic of styrene and PDPMA.

The broad peak observed at δ 7–7.4 ppm (a) corresponds to the phenyl ring protons. To determine the composition, the peak (b) at δ 2.5–2.6 ppm due to the protons α to the phenyl ring of PDPMA (Ph-**CH_2_**-(CH_2_)_13_-CH_3_) was selected as the characteristic peak of PDPMA. Other peaks, (c) the hydrogen on carbon β to the aromatic ring (Ph‒CH_2_‒CH**_2_**‒) was seen at δ 2.2 ppm and (d), broad peak) the ‒CH_2_‒ and ‒CH‒ protons of the main chain were observed at δ 1.6‒1.43 ppm. The internal methylene protons of the side chain of PDPMA appeared at δ 1.2 ppm (e), and the peak at δ 0.83 ppm(f) was due to the terminal -CH_3_ protons of the alkyl side chain. Look at the composition data determined from ^1^H NMR and the feed composition tabulated in Table 1, it can be seen that the copolymers had lower PDPMA content than the feed. The F-R and K-T parameters, for determination of the reactivity ratio (r_1_, r_2_), were calculated from the composition data and the values are tabulated in Table 2 and corresponding plot is shown as Figure 3.

The r_1_, r_2_ values determined from the plots were tabulated in Table 3.

The reactivity ratio for styrene (r_1_) was higher than that of PDPMA (r_2_) and the product r_1_ r_2_ was close to zero (the value is between zero and 1). The values suggested random copolymerization and the chances for incorporation of styrene was more with a tendency to alternate, as the product r_1_ × r_2_ was less than 1 and close to zero [46]. If the average of r_1_ and r_2_ obtained from F-R and K-T methods were taken, then r_1_ = 0.93 and r_2_ = 0.05. The values imply that styrene-type radical S* shows slightly more preference for copolymerization than for homopolymerization. The r_2_ value suggests PDPMA has very less preference for homopolymerization but prefer copolymerization with styrene monomer. Thus, styrene enters the copolymer faster than PDPMA, since it has equal chance of copolymerization and homopolymerization resulting in long sequences of sty units throughout the whole composition range studied.

Basing on the reactivity ratio, the sequence length was determined and the values, given in Table 4, suggest higher styrene sequence length (I_1_) with high styrene content in the feed. The PDPMA sequence length (I_2_) remained more or less the same, in agreement with its alternating tendency, or inability to form long sequences of PDPMA units.

The composition data, determined from ^1^H NMR given in Table 1, also supports this observation as the PDPMA content in copolymer was low, despite the increase in feed concentration.

The observed value of reactivity ratio was different from that reported in the literature for copolymerization of styrene and hydrogenated cardanyl acrylate or PDPA [38]. They reported r_1_ = 0.34 for styrene and r_2_ = 0.97 for PDPA [38]. The observed difference shows the methacrylate has low propensity for homopolymerization in presence of styrene than PDP acrylate. The difference might also be due to the different initiating system employed; as the literature studies were performed using cobalt based initiating system [38]. Similar observation of initiating system influencing the reactivity was also observed by Koiry and Singha in the polymerization of styrene with 2-ethyl hexyl acrylate [47]. They suggested a difference in the polymerization mechanism of ATRP and conventional radical polymerization due to the difficulty in formation of the polar catalyst complex with monomers having non-polar side chain. This would be applicable to the present case as well, since the monomer has a 15 carbon alkyl side chain to impart hydrophobicity.

### 3.2. Molecular Characterization of Styrene-PDPMA Copolymers

The molecular characteristics of the synthesized copolymers were studied using SEC and the values are tabulated in Table 1. In all cases, except for styrene homopolymerization, deviation from the theoretically calculated molecular weight was seen. The difference increases with increasing the PDPMA content of the feed. The difference might be attributed to the variation in hydrodynamic volume of SPDPMA copolymers resulting from the 15 carbon alkyl chain present in PDPMA. The alkyl chain on each repeat unit would act like short branches and branched polymers have a compact structure, thus smaller hydrodynamic radius and consequently have lower molecular weight. The relation between branching, hydrodynamic radius and intrinsic viscosity or molecular weight had been previously reported [48]. Similar observations were also reported in the case of styrene-ethyl hexyl acrylate copolymers synthesized by ATRP [33].

The variations in molecular weight distribution of the different copolymers with increasing PDPMA concentration are shown in Figure 4. The polydispersity values obtained from GPC were also found to vary with increasing PDPMA content in the copolymer.

The polystyrene homopolymer showed the narrowest distribution and with increasing PDPMA content, the polydispersity increased, suggesting loss of control over the polymerization or indications of a branched structure with increasing PDPMA content.

### 3.3. Glass Transition Behaviour of Styrene-PDPMA Copolymers

The glass transition behaviour of representative styrene-PDPMA copolymers studied by DSC are shown in Figure 5.

The homopolymer (PDPMA100) showed a small endotherm at about 1 °C. At 10 wt % PDPMA content. Base line shift characteristic of the T_g_ of amorphous polymer was seen at about 10 °C and a crystalline melting like transition at about 47 °C. At 30 wt % PDPMA the endothermic melting like transition became clearer. The copolymer with 50 wt % PDPMA content showed a broad glass transition, centered at about 3.85 °C but no clear endothermic, melting peak was visible, though we expected a better side chain melting transition in this case due to the higher PDPMA content. However, as expected, with increasing PDPMA concentration in the copolymer the glass transition temperature (T_g_) was found to be decreasing. The melting transitions, other than glass transition were earlier observed in comb-like polymers of n-alkyl acrylate polymers, due to the crystallizable side chains, as in the case of PDPMA. The melting transitions are characteristic of side chain crystallisable polymers and the T_g_ mostly overlaps with the melting transition [18,37,49].

With 10 wt % PDPMA mostly a softening effect was observed, with 30% PDPMA the crowding of alkyl chains lead to better crystallization, and with 50 wt % PDPMA, the crystallization appears to be not favoured, though the present studies, could not provide any insight for this observation. Sidney et al. described an investigation of the side chain crystallinity found in polymethacrylates and polyacrylates with n-alkyl groups from 12 to 18 carbon atoms in length as the similar way. These crystallites are made up of the n-alkyl groups which extend from the backbone of the molecules, rather than of segments of the backbone itself [50]. The overlap/crowding or interdigitation of alkyl chains on different polymer chains are shown here. If crystallization is favored in the copolymer then a two-phase structure resembling packed alkyl crystalline domains (LHS of Scheme 4) distributed in the amorphous matrix is formed. On the RHS of Scheme 4, the interdigitation of alkyl chains on two different chains is shown. The observed phase separation/crowding of alkyl chains in SPDPMA30, might be presented schematically as in Scheme 4 below.

The formation of this type of a phase separated domain structure was further confirmed using transmission electron microscopy. The TEM images of SPDPMA70 shown in Figure 6, confirms the formation of a phase separated, core-shell type structure with crystalline (light color) domains and aromatic, electron rich styrene domains.

### 3.4. Thermal Stability of Copolymers

In order to assess the thermal stability of the copolymers, thermogravimetric analysis (TGA) was carried out in the temperature range from −50 °C to 200 °C at 10 °C min^−1^.Typical plots are shown in Figure 7.

Copolymers with high (90 and 70 wt %) styrene content displayed the maximal thermal stability. The derivative thermogram of the copolymer sample with lower styrene content (30 wt %) (Figure 8B) showed two step decomposition, at 265 °C and 435 °C, whereas the sample with 50 and 70 wt % styrene content decomposed in a single step. The reduction in thermal stability at high PDPMA concentration might be attributed to polar carbonyl groups, increasing its susceptibility to degradation whereas at higher styrene content the thermal stability of the phenyl group contributes.

### 3.5. Rheological Properties

The rheological properties were measured in oscillatory shear mode as a function of frequency, to study the changes in storage modulus (G′), loss modulus (G″) and complex viscosity (η*). The analysis was carried out at different temperatures with a strain rate of 0.5%. Typical data are shown in Figure 8.

With increasing PDPMA content in the feed, the modulus and complex viscosity was decreasing. The copolymer with 70 wt % PDPMA content showed very low modulus at elevated temperature and so the maximum measurement temperature was 80 °C but 130 °C for the copolymer with 30 wt % PDPMA content. In other words, on increasing styrene content the viscoelastic properties were better retained at elevated temperature and the data could be measured at 130 °C as well. On similar grounds, the minimum temperature was 50 °C for the sample with 70 wt % styrene content, since the high styrene resin was hard at lower temperatures to facilitate data collection. Probably, such samples with high styrene content might be useful for hot melt adhesives.

### 3.6. Adhesion Behavior

The adhesion properties, namely lap shear strength (LSS) and peel strength (PS), were evaluated on different substrates as per ASTM standards D1002-10 and D903-98, respectively. The copolymers with 70, 50, and 30 wt % styrene content were tested. Test specimens were prepared using Kraft paper and polyethyleneterepthalate (PET). The sample with 30 wt % styrene content showed negligible adhesion strength, irrespective of the substrate and so could not be tested. The results obtained for 70 wt % and 50 wt % styrene content copolymers were summarized in Table 5 below.

As seen from Table 5, the LSS on paper substrate decreased from 393,000 Pa to nearly half the value, and the peel strength decreased by six times, as the styrene content was decreased from 70 to 50 wt %. On the mixed substrate (paper-PET), the LSS was 367,000 Pa at 70 wt % styrene content and only a marginal decrease was observed at 50 wt % styrene content whereas the peel strength values decreased by nearly half. Thus, the maximum lap shear strength values were obtained on both paper-paper and on mixed substrate (paper-PET) at 70 wt % styrene content. The maximum peel strength was obtained on paper substrate at 70 wt % styrene content and the value reduced with 50 wt % styrene content without any appreciable variation on other substrate. In all cases, the lap shear strength was higher in comparison to the peel strength, for the 70 wt % styrene content copolymer. To summarize, the variation in adhesion properties with composition suggests tuneability of the adhesion properties through copolymerization of styrene and PDPMA.

## 4. Conclusions

In this work the copolymerization behavior of styrene and pentadecyl phenyl methacrylate (PDPMA), at different monomer feed ratio was studied using atom transfer radical polymerisation technique. The copolymer composition and reactivity ratios were calculated following Finemann–Ross and Kelen-Tudos methods, using the composition data obtained from ^1^H NMR spectroscopy. The values suggest preferential copolymerization of styrene with alternation tendency and the formed copolymer was rich in styrene at any feed composition, despite increasing the PDPMA content in feed. Sequence length determination based on the reactivity ratio values suggest more less constant sequence length of PDPMA and decreasing styrene sequence length with increasing PDPMA content in feed. Molecular characterization of the copolymers by GPC reveals dependence of molecular weight and polydispersity on the feed composition. The copolymer molecular weight varied between 3000–32,000 g/mole and the polydispersity was in the range 1.32–2.19, depending upon the feed composition. The glass transition behavior of the copolymers varied with the feed composition and the thermal stability of the copolymer was also high with increasing styrene content. The morphological characterization by TEM suggested a phase separated morphology. Rheological and adhesion tests suggest tuneability of the adhesion properties through copolymerization of styrene and PDPMA and their utility in hotmelt adhesives.

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
