# Peer review of "Copolymerization of Styrene and Pentadecylphenylmethacrylate (PDPMA): Synthesis, Characterization, Thermomechanical and Adhesion Properties"

_polymers, 2020, doi:10.3390/polym12010097_

Round 1
Reviewer 1 Report
This paper reported the copolymerization behavior of PDMPMA with styrene under the typical ATRP condition. The authors successfully determined the reactivity ratio for this system by two different methods. In addition, the thermomechanical and adhesion properties of the copolymers were also studied in detail. The scientific results described in this paper are detailed and clear. I think that this paper is worth publishing in Polymers. However, some minor revisions would be beneficial to the current manuscript:
Many of the figures are hard to look, especially the TOC, Figures 4, 5, 6, 7, and 8. Those figures should be replaced with their higher resolution version. The chemical structure drawing of the styrene-PDPMA copolymer (in the TOC, Scheme III, and Figure 2) may lead misunderstanding and therefore need to be revised. From the experimental results, the obtained copolymers should have random sequence of styrene and PDPMA. However, the chemical structure indicates the block copolymer-type sequence. The reviewer could not understand what the left side of Scheme IV means. More explanation is needed.
Author Response
Reviewer 1.
Many of the figures are hard to look, especially the TOC, Figures 4, 5, 6, 7, and 8. Those figures should be replaced with their higher resolution version. The chemical structure drawing of the styrene-PDPMA copolymer (in the TOC, Scheme III, and Figure 2) may lead misunderstanding and therefore need to be revised. From the experimental results, the obtained copolymers should have random sequence of styrene and PDPMA. However, the chemical structure indicates the block copolymer-type sequence. The reviewer could not understand what the left side of Scheme IV means. More explanation is needed.
We thank the reviewer for appreciating the work, and clarity of presentations.
The figures are improved and presented now in higher resolution.
The chemical structure drawing of the styrene-PDPMA (Scheme III and Fig 2) was drawn more precisely
We agree that the sequence of styrene & PDPMA is random. Accordingly, the chemical structure has been revised in TOC, FIG2 and Scheme 3. The hypothetical domain structure is shown on the left side of Scheme IV. It results from the lamellar ordering of alkyl chains. The right site of Scheme IV shows enlarged view of one such domain with interdigitating alkyl chains. The discussion is also modified to make this point clear.
Reviewer 2 Report
The manuscript under consideration by Joseph et al is a study related to copolymerization of St and PDPMA by ATRP. The copolymerization reactivity ratios were calculated by NMR data using KL and FR methods. The thermal, rheological and adhesion behavior of synthesized copolymers were evaluated and correlated to its composition. In general, paper is well-written and presented in a logical manner. However, I have some suggestions/concerns that should be addressed before acceptance of this paper in Polymers.
The introduction must be improved with more citations and to build a strong justification of importance of this study I wonder why authors have used PI SEC standards for correlation to the molar mass. Authors have also mentioned at some point that even different architecture same polymer may result in different elution time and in turn the incorrect molar mass obtained. I would strongly recommend to use PS standards for their calculations that should give more realistic values closest to real. In case of PI standards, there are two sources of errors, first is different composition and second is architecture. I am sure, the values obtained by using PS standard calibration curve will be more closer to the theoretical values. The real SEC chromatograms should also be included Furthermore, ATRP always give narrow distributions when conversions are kept low. With near to 100% conversions, the dispersity values are bound to increase Another very important point is to have all the samples with similar molar mass including both PS and PPDPMA homopolymers along with the copolymers with varying compositions. Then compare and show the shift of all the tested properties from one homopolymer to other hompolymer going through a composition change. This will ensure effect of composition on these properties. To elaborate further, Figure 4-8 should also have hompolymers of both types with similar molar masses of all the products. This will ensure composition selectivity of the samples for all the properties.
Author Response
The introduction must be improved with more citations and to build a strong justification of importance of this study I wonder why authors have used PI SEC standards for correlation to the molar mass. Authors have also mentioned at some point that even different architecture same polymer may result in different elution time and in turn the incorrect molar mass obtained. I would strongly recommend to use PS standards for their calculations that should give more realistic values closest to real. In case of PI standards, there are two sources of errors, first is different composition and second is architecture. I am sure, the values obtained by using PS standard calibration curve will be more closer to the theoretical values. The real SEC chromatograms should also be included Furthermore, ATRP always give narrow distributions when conversions are kept low. With near to 100% conversions, the dispersity values are bound to increase Another very important point is to have all the samples with similar molar mass including both PS and PPDPMA homopolymers along with the copolymers with varying compositions. Then compare and show the shift of all the tested properties from one homopolymer to other homopolymer going through a composition change. This will ensure effect of composition on these properties. To elaborate further, Figure 4-8 should also have homopolymers of both types with similar molar masses of all the products. This will ensure composition selectivity of the samples for all the properties.
Thanks the reviewer for appreciating the general presentation of results.
Introduction - We added new references to highlight the importance of development of sustainable products using biobased materials and its structure-property correlation study.e.g
We have used PI SEC standards as the methyl group on isoprene units resembles short branches and hence it is similar to the copolymer structure with alkyl chain branches, furthermore the backbone of PI is more flexible than in case of PS and in our opinion the choice of PI standard was justified. It is agreed that PS standards will give different result and we will make this comparison in the future studies.
In our work we found it difficult to prepare homopolymer and copolymer of identical mol wt due to the different reactivity of the monomers. The homopolymerization of PDPMA was observed to be very fast, such that in about 30 minute 70% conversion was achieved. Nevertheless, we are working to see if other catalyst system/ligand can be used to achieve identical mol and will be the subject matter of future communication.
The ideal situation to have homopolymer of identical molar mass was found to be difficult in our experiments. This aspect we will bear in mind in our future studies to study the composition selectivity of the properties. From our results the property-structure relationship could not be exactly quantified but its existence was clearly shown.
To include both homopolymer properties in the figures 4-8 estimated using the same procedures as for investigated copolymers seems now not possible because this part of experimental work is finished at MG University and further experiments are performed by the same team at Gdansk University of Technology.
Round 2
Reviewer 2 Report
Authors adequately addressed the concerns and provided justifications for the points they are unable to address. I recommend publication in current form.
Author Response
Dear Reviewer,
Thank you very much for the in-depth review that will help us to write our next publications better
Sincerely,
The Authors